# Factors to determine the adoption of online teaching in Tanzania's Universities during the COVID-19 pandemic

Mackfallen G. Anasel[1]*, Idda L. Swai[2]

**1** School of Public Administration and Management, Department of Health Systems Management Mzumbe University, Morogoro, Tanzania, **2** School of Public Administration and Management, Department of Local Government Management, Mzumbe University, Morogoro, Tanzania

* mganasel@mzumbe.ac.tz, maremay2k@yahoo.co.uk

## Abstract

### Background

Face to face mode of delivery has been a standard method of teaching courses in the majority of African Universities Tanzania included. The COVID-19 pandemic has caused the closure of all schools and universities worldwide; therefore, face-to-face teaching is no longer the only appropriate and feasible teaching method. This requires changes in the teaching method with the remarkable rise of e-learning, whereby teaching must be undertaken remotely and on digital platforms.

### Objective

The main objective of this study is to determine the perceived factors that hinder the adoption of online teaching in Tanzanian universities.

### Method

A mixed method dominated by a quantitative approach was used to answer the research questions. A total of 173 academic and ICT staff from nineteen universities in Tanzania participated in this study. Descriptive statistics (univariate analysis) and binary logistic regression were applied to analyse the frequency and compare the mean to describe the characteristics of respondents and determine the factors influencing the academic staff to have an online course. This was followed by content analysis to analyse the academic staff's proposed adaptation to online classes.

### Results

The findings indicated that the main hindrances to online teaching are lack of capacity; knowledge on how to conduct online courses; knowledge and technological factors; environmental factors; staff attitudes towards delivering online courses, and years spent using eLearning. The academic staff proposed improving ICT infrastructure and strengthening the capacity of academic staff to provide online courses.

**Data Availability Statement:** The manuscript and its Supporting information files contain a minimal data set has been uploaded. The data set is in SPSS format and anonymised to avoid identifying the respondents and the Institution.

**Funding:** the authors received no specific funding for this work.

**Competing interests:** The authors have declared that no competing interests exist.

## Conclusion

The study concludes that most academic staff are willing to adopt online teaching while suggesting improving the organisational and individual factors to enhance online education. The paper recommends that the university use freely available online teaching tools and platforms while simultaneously focusing on addressing the corporate and individual factors identified in this paper to enhance online teaching, which is mandatory in response to lockdown measures.

## 1.0 Introduction

On March 17, 2020, the Tanzania government suspended all political gatherings and closed all schools, colleges and universities for thirty days. The suspension was extended for an indefinite period by Prime Minister Kassim Majaliwa on April 14 2020 [1]. Later the late President John Pombe Magufuli announced the opening of all institutions on 1[st] June 2020. These measures have been done to respond to the declaration made by the World Health Organization on March 11, 2020, that COVID-19 is a global pandemic disease and, therefore, adherence to the prevention and control measures was mandatory [2]. During the first announcement, most universities in Tanzania were mid-recess after first semester examinations, per the university almanac.

The UNESCO report shows that 9.8 million African students stay home without continuing their studies after the closure of universities [3]. As there is uncertainty about the duration of this pandemic and the requirement of social distancing measures for long, online teaching and learning have become a necessity for education around the globe during the pandemic [4] and therefore, the universities are forced to take prompt measures, including a move to online courses teaching. However, the movement has encountered several challenges, including internet and power supply access. Only 24% of the population in Africa has access to the internet, which has poor connectivity, is costly and is frequently interrupted by power cuts [5]. This has led to the use of smart devices, laptops and power banks for power backup. In addition, limitations to online learning are evident. Among them are start-up funding for investment in online learning, low literacy lack of preparedness among lecturers and students and lack of timely feedback when the students post questions in the discussion forum [4, 6]. Most Universities in Tanzania rely on the face-to-face mode of delivery thus creating difficulties for the universities to change dramatically to offering online courses. In addition, most programmes were not designed to be offered online.

Several studies have indicated that good Information Communication Technology (ICT) Infrastructures are considered an essential element when planning for an online course [7–11]. This requires investment in ICT infrastructures and capacity building using ICT in the classrooms while allowing students to follow sessions and learn outside of the school environment. The study by [12] indicates that Tanzania has stable and fast-growing broadband network services of 3G and 4G. The paper [13] further shows the tendency to lower the internet bundle due to competition, affecting the internet speed. Another study [14] indicated that more than 50% of respondents use Internet services for searching news, games, and entertainment, while only 12.5% use the Internet for academic purposes. This may pose several questions and further requires examining the commitment and readiness of learners to online learning. The Universities in Tanzania have different ICT infrastructure levels, equipment and human resources capacities for organising and delivering online courses. Online course

delivery facilitates trust, rapport and collaboration to enhance the learning experience if well developed and utilised. For instance, the University of Dar es Salaam offers three blended learning programs: a Postgraduate Diploma in Education (PGDE), a Postgraduate Diploma in Engineering Management (PGDEM) and a Master's degree in Engineering Management (MEM) at regional centres in Mbeya, Mwanza, Dar es Salaam, and Arusha via Moodle, with limited-few face-to-face meetings. The Open University of Tanzania also uses Moodle to offer its programs via 28 regional centres in Tanzania. Mzumbe University was offering a Master of Business Administration, Master of Science in Accountancy and Finance, Master of Science in Procurement, and Master of Science in Project Planning and Management in a modular form at Mwanza Centre [13].

Given the above, the paper investigates factors that influence the adoption of online courses as an alternative to course delivery and teaching during an outbreak of COVID-19 and other pandemics such as Marburg Virus Disease (MVD) that was confirmed on March 2023 in the north-west Kagera, Tanzania and requires adhering to prevention and control measures that may include social distancing and lockdown in severe cases. While delivery of courses online is optional in most of the universities in Tanzania, adopting online courses is also an alternative to limited teaching infrastructures such as lecture halls and classrooms where the students can follow their systems at home or in their hostels. Another potential benefit from the learners' perspectives is that online learning is self-regulated, where learners tend to use various cognitive and metacognitive strategies to accomplish their learning goals [15]. Online learning has contributed to expanding access, improving the quality of learning, preparing students for a knowledge-based society and gaining e-autonomous study skills while expanding learning opportunities and profit-making with little use of resources [4, 16].

This study has been guided by three main questions, (1) What are the factors that influence the university staff towards shifting to online teaching and (2) What are the perceived obstacles that will hinder the universities from shifting to online teaching (3) What are the proposed adaptation to online teaching by the academic staff.

Various studies have outlined factors determining the adoption of online courses in higher learning institutions, as illustrated in Fig 1. The factors include technological [14, 17–19], organisational [14, 19, 20], environmental [17, 18, 22] and nature of the courses [20–22]. This study has included staff attitudes towards online teaching and reasons that make staff think twice before attending online classes. The study starts with an analysis of how these factors influence the adoption of online course delivery and is followed by the characteristics that determine the willingness of staff to teach online courses during the pandemic or as a way to use varying learning styles and different technologies in teaching in higher learning institutions.

## 2. Methods

### 2.1 Study approach

A quantitative study design mixed with qualitative research methods was performed to achieve the research objectives. While the quantitative approach seeks to build on directly observable quantitative indicators and establish causal relationships, the qualitative approach, owing to data collection and analysis flexibility, can necessarily be adjusted as the research proceeds [23]. These approaches were meant to complement each other, prove the existing theories on adopting online courses through quantitative methods, and explore staff opinions and recommendations on adapting online teaching through a qualitative approach.

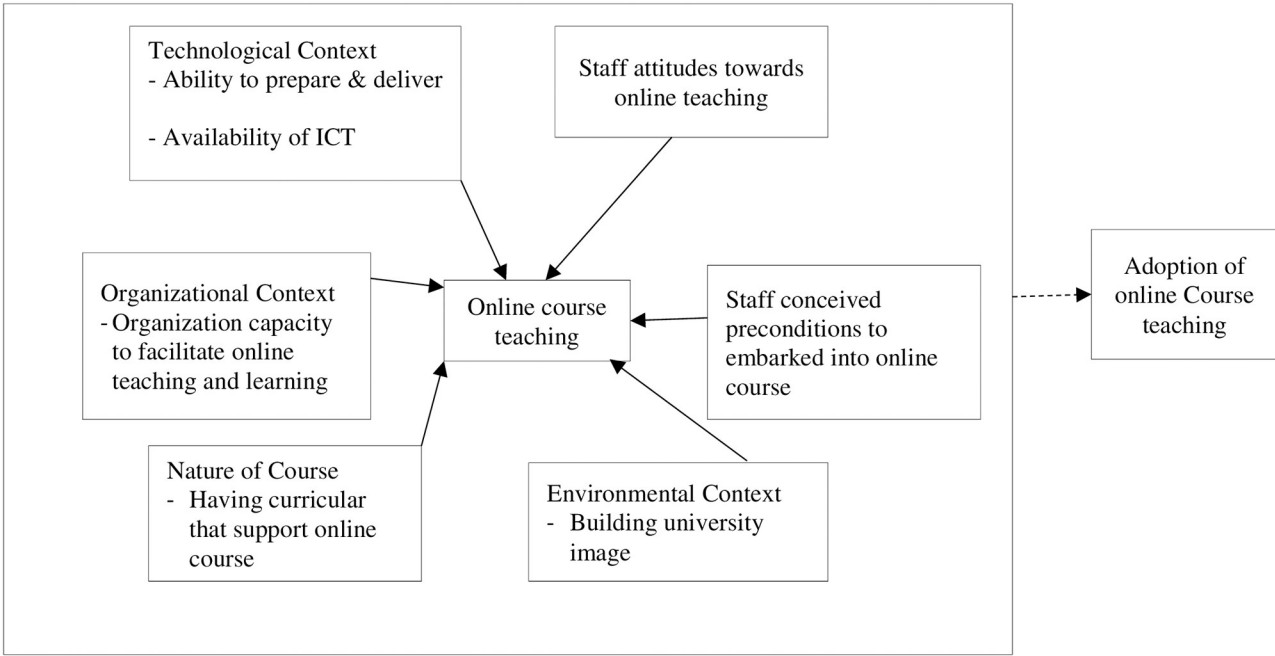

**Fig 1. Factors determine the adoption of online teaching.**

## 2.2 Sampling and data collection

The snowball technique was used to reach the participants. Google Form was used to collect quantitative and qualitative data following the advice health professionals and the government gave regarding limiting unnecessary movement during the COVID-19 pandemic. The individuals with their email addresses and telephone numbers were identified and communicated with them to introduce the study. They requested them to fill out the questionnaire and circulate the link within their network. Of the 14 individuals consulted, 12 agreed to fill out the Google form and ultimately share the same within their network, and two needed a smartphone and more time to be ready to accept our request. We also emailed the Directorate of Research and Publication of Five Universities *(withheld names)*, requesting their office to introduce our study and share the link with their staff. Three universities agreed to share the link with their team, but two Universities have yet to respond to our email. Finally, 173 respondents from nineteen (19) universities participated in the study. The Universities were Jordan University College, Kampala International University in Tanzania, The Muhimbili University of Health and Allied Sciences, Mwalimu Julius K. Nyerere University of Agriculture and Technology, Mwenge Catholic University, Mzumbe University, National Institute of Transport, Nelson Mandela African Institution of Science and Technology, Ruaha Catholic University, Saint John's University, St. Augustine University of Tanzania, Sokoine University of Agriculture, Tanzania institute of accountancy, Tanzania Institute of Accountancy, The Open University of Tanzania, The State University of Zanzibar, The University of Dodoma, United African University of Tanzania and University of Dar es Salaam.

The quantitative and qualitative questions list was self-developed based on theoretical reviews and the conceptual model in Fig 1. The questions were grouped into three categories. The first category seeks to collect the demographic particulars of the participants, including the name and the nature of the university, the profession and experience in teaching. The

second category included closed-ended nature questions that aimed at collecting quantitative data. In contrast, the third category was the open questions intended to collect individual opinions, experiences and recommendations on adopting online courses. Therefore, the same Google form collected both qualitative and quantitative data.

## 2.3 Data management and analysis

Data management and cleaning was the first research process done after transferring the data set from the Google form to an Excel sheet. Data management involves creating, organising, editing and coding the data. There were one hundred and seventy-seven (177) respondents. Four respondents were omitted from the analysis due to various factors. Two respondents did not respond to the concert statement, one responded to the concert statement but did not provide any information, and one was from Virginia Commonwealth University in the USA. Although the respondent from Virginia University provided relevant and helpful information related to online courses in Tanzania, it was omitted from the analysis because the study focuses on academics working in Tanzania Universities. In addition, two questions required the respondents to mention issues in their E-learning Platform and the reasons that make them think twice before deciding to use an online course. The responses were further coded to reflect all details of the answers given by the respondents. The codebook was created, sorted into categories and groups, entered into Excel, cleaned and exported to SPSS for data transformation and analysis.

The quantitative data analysis was divided into descriptive statistics (univariate analysis) and binary logistic regression. Descriptive statistics were applied to analyse the frequency and compare the mean to describe the characteristics of respondents and identify the items with a high mean. This was followed by regression analysis to determine the factors influencing the academic staff to have an online course. Having an online course or not follows under dichotomous outcome that calls upon the uses of logit [24]. The model is described in the following logit function below.

$$Logit(y) = \beta_0 + \beta_1 X_1 + \beta_2 X_2 \ldots . \beta_p X_p$$

Where;
Y is a dichotomous dependent variable called **logit,** defined as;
**y** = (1 = having online course; 0 = not having online course)
$\beta_0$ = Is the intercept
$\beta_1, \beta_2, \beta_P$ = Logistic regression coefficient of $X_1, X_2, X_3$ respectively
$X_1, X_2, X_3$ = Independent variables

Before running logit regression, we checked for the scales' internal consistency using Cronbach's alpha test. The Cronbach's alpha test was done to ensure that all variables transformed measures the same underlying construct in Table 1 (*see* S1 Appendix). The Alpha coefficient values range from 1 to 5, describing the reliability of factors extracted from Likert scale variables. The higher the score, the more reliable the generated scale: the 0.7 (70%) values indicate an acceptable reliability coefficient [25, 26]. As shown in Table 1, Cronbach's alpha for staff knowledge, technological factor, organisational factor, environmental factor, nature of the course, and staff attitudes towards an online course is above 70%, showing that the selected items measured the same underlying construct. It should be noted that the items were removed one by one, as indicated in Appendix 1, to ensure that the remaining items measure the same underlying construct.

The content analysis was used to analyse the academic staff's proposed adaptation to online courses. The analysis started with repeatedly reading the explanations provided by the staff to

**Table 1. Reliability statistics for factors that determine online courses teaching.**

| S/N | Context | Number of Items | Cronbach's Alpha |
|---|---|---|---|
| 1 | Knowledge & Technological factors | 12 | 88.9 |
| 2 | Organizational Factors | 6 | 77.1 |
| 3 | Environmental factors | 4 | 88.8 |
| 4 | Nature of the course | 2* | 74.7 |
| 5 | Staff attitudes towards online courses | 7** | 84.2 |

Key:

* Three items were deleted

** Three item was deleted

cross-check the data's quality and acquire an overall sense of the data. Some themes were pre-determined when reviewing the literature and designing the tool for data collection. Other themes were captured deductively when reviewing the narration noted down by respondents. Similar themes were merged, meaning attached and used to conclude the qualitative section.

## 2.4 Ethical consideration and consent

Approval to conduct this study was obtained from the Postgraduate Technical Committee, Mzumbe University, before the commencement of the study. Since the survey was conducted during partial lockdown (closer of all schools, Colleagues and Universities), the Google form has opted for data collection to restrict unnecessary travel and movement for researchers and ultimately adhere to prevention and control measures. Therefore, the link to the Google form was also circulated to the Directorate of Postgraduate Studies in the universities, where data was collected for their review and requested to circulate the link to the study participants in the universities. The form starts with the consent statement, where all respondents were required to indicate their agreement to participate in the study. The statement allowed the participants to continue or quit at any time, and there were no forms of payments or rewards associated with her/his participation. The participants were required to tick the *yes* box to indicate their consent to participate in the study. All participants were employees of the universities where the data was collected; therefore, no minor was involved in the study. Therefore, the data in this paper are from the respondents who agreed with the consent statement after reading and understanding the purpose of the study. Comparative analysis across institutions was not done to hide the identity of the participants and their institutions.

## 3.0 Results

### 3.1 Characteristics of respondents

A total of 173 academic staff from twelve Universities participated in this study. While 76 (44%) respondents hold positions such as heads of department and principal/dean/director, the majority of respondents (97(56%) do not hold any managerial position in their universities. More than three-thirds of respondents, 137(79%), are from public Universities, and the rest are from private universities. In addition, the respondents dominated men; that is, more than half of all respondents were men indicating the under-representation of women respondents in the study. 18(10%) participants are information technology experts, including the ICT officers and instructional designers whose role is to support lecturers and students on eLearning issues. Half of the respondents have an experience of more than ten years, and the majority of them have master's degrees and a PhD. Only 37(21%) respondents reported having an

online course. However, the majority of the respondents, 153 (88%), are willing to teach an online course, of which about 54% know related to online courses and teaching, as indicated in Table 2.

## 3.2 Materials included in the e-learning platform

The study required the respondents who have an online course to indicate the content of what is uploaded online. The content list was prepared after consultation with e-learning and online teaching experts, and the respondents were required to tick the uploaded content. The findings showed that the content uploaded to the E-learning platform by most participants was dominated by teaching slides, course outlines, and journal articles used in teaching. Video clips, discussion forums and tests were the least materials reported to be uploaded by some participants in the E-learning platform, as indicated in Table 3.

## 3.3 Prospects that make academician hesitate before embarking on online teaching

Table 4 provides reasons why the staff hesitate before starting online course teaching. The study results indicate a technical problem; the effectiveness of online courses and difficulties involved in the online assessment are the factors with high mean. This means that most respondents pointed to these factors as the issues they considered before going to online teaching. The study findings showed that fear of status and role, fear of being a victim of fashion and fear of losing recognition are the factors that are less concern when thinking about embarking on online teachings.

## 3.4 Obstacles that hinder the universities shifting to online courses teaching

The study explores potential obstacles that are likely to hinder universities from shifting to online teaching and learning. Table 5 shows the variables that are included in the final logit model. All participants' characteristics described in Table 2 and determinate factors for online teaching in Table 1 were entered in SPSS software using enter method. The variable with the highest p-value was removed first from the model, and the model fit and significant level were rechecked until they remained with the significant variables for the final model.

The findings in Table 5 show that academic staff qualification; knowledge of administering online courses; knowledge and technological factors, environmental factors, staff attitudes towards delivering online courses and years spent in using E-learning have significantly influenced the delivery of online courses. For example, a staff with a master's degree has a likelihood of 10.33 to deliver online courses compared considerably to staff with a bachelor, diploma and PhD. Having a master's degree; knowledge of how to provide online courses; environmental factors; and knowledge and technological factors were indicated to influence the decision to teach online courses.

Most respondents were concerned about the capability of higher learning institutions in Tanzania to administer online courses and teaching. The participants mentioned issues such as limited internet connection, lack of staff capacity related to online teaching and poor infrastructure to support online teaching, such as video conferencing. One of the respondents shared that;

> *"Most universities in Tanzania, as in other countries in the developing world, do not have infrastructures to support online teaching. For example, there is a big problem with an*

**Table 2. Respondents' characteristics (N173).**

| Characteristics | Frequency | Percent |
|---|---|---|
| Position at University | | |
| None | 97 | 56.1 |
| Head of Section/Department | 68 | 39.3 |
| Principal/Dean/Director | 8 | 4.6 |
| Academic Rank | | |
| Supporting Staff | 18 | 10.4 |
| Assistant Lecturer | 63 | 36.4 |
| Lecturer | 63 | 36.4 |
| Senior Academic Staff | 29 | 16.8 |
| College of affiliation | | |
| Applied Sciences | 105 | 60.7 |
| ICT/Education | 22 | 12.7 |
| Social Science | 46 | 26.6 |
| Nature of University | | |
| Private | 36 | 20.8 |
| Public | 137 | 79.2 |
| Sex | | |
| Female | 41 | 23.7 |
| Male | 132 | 76.3 |
| Age | | |
| 25–35 years | 52 | 30.1 |
| 36–45 years | 77 | 44.5 |
| 46–60 years | 40 | 23.1 |
| Above 61 | 4 | 2.3 |
| Working Experience | | |
| Less than one year | 4 | 2.3 |
| Between two and five years | 32 | 18.5 |
| Six to ten years | 50 | 28.9 |
| Eleven years and above | 87 | 50.3 |
| Highest Academic Qualification | | |
| Diploma | 1 | 0.6 |
| Bachelor | 8 | 4.6 |
| Master | 86 | 49.7 |
| PhD | 78 | 45.1 |
| Having Online Course | | |
| No | 136 | 78.6 |
| Yes | 37 | 21.4 |
| Number of Course Administered online | | |
| 0 | 138 | 79.8 |
| 1–2 | 26 | 15 |
| 3–5 | 9 | 5.2 |
| Year spent in E-learning | | |
| Never Use E-learning | 75 | 43.4 |
| 1–5 years | 74 | 42.8 |
| Above six years | 24 | 13.9 |
| Ever Study Online Course | | |
| No | 101 | 58.4 |

(*Continued*)

**Table 2.** (Continued)

| Characteristics | Frequency | Percent |
|---|---|---|
| Yes | 72 | 41.6 |
| Having Knowledge on how to run online course | | |
| No | 80 | 46.2 |
| Yes | 93 | 53.8 |
| Willing to teach online course | | |
| No | 20 | 11.6 |
| Yes | 153 | 88.4 |

*internet connection–strength, i.e., weak every time; lack of ICT facilities, e.g., computer lab with functioning computers; lack of dedicated lecture rooms for video conferencing".*

Respondent 6

This was further explained;

*"Effectiveness of online courses depends more on the state of the country. For instance, many rural areas in Tanzania are disconnected from effective internet. This is a barrier, even if Universities have all facilities required. . .Further, some courses have practical exercises that can never be replaced by online teaching."*

Respondent 137

Another issue raised by academic staff is the difference in the economic status of the students. Most respondents believed that most students in higher learning institutions need the necessary devices, such as laptops and smartphones, that they can use to follow courses for online learning. Another issue that looms most is the accessibility and affordability of the internet. The internet connection, particularly in rural areas where most Tanzanians live, could be better, so before we raise concerns on whether the students can afford the cost related to the internet, we need to see whether the internet can be accessed and at what speed.

**Table 3. The content that is uploaded to e-learning platforms.**

| Content | Mean |
|---|---|
| Slides (Teaching Materials | 0.42 |
| Course Outline | 0.42 |
| Journal Articles | 0.35 |
| Course Introduction | 0.34 |
| Books | 0.34 |
| Seminar Questions | 0.33 |
| Learning Outcomes | 0.33 |
| Quiz | 0.26 |
| Study Cases | 0.25 |
| Announcement | 0.25 |
| Test | 0.24 |
| Discussion Forum | 0.24 |
| Video Clip | 0.22 |

**Table 4. Reasons that make staff to think twice before embarking into online teaching.**

| Reasons | Mean |
|---|---|
| Technical problems | 0.66 |
| Effectives of online to replace traditional lectures | 0.48 |
| Difficulties involved in online assessment | 0.48 |
| Fear to lower learning level for student | 0.47 |
| Difficulty dealing with online interaction | 0.41 |
| Risk of decreasing the richness of interaction | 0.39 |
| Adequacy of payment for time spent online | 0.29 |
| Control of time spent in online interaction | 0.29 |
| Difficulty achieving quality design | 0.28 |
| Fear of having a less effective course | 0.25 |
| Difficulty of moving on to a learners-centered approach | 0.24 |
| Lack of time to prepare online content and activities | 0.23 |
| Appropriation of content development by other people | 0.20 |
| Fear of students' rejection | 0.18 |
| Fear about legal issues | 0.12 |
| Lack of understanding of institutional vision and strategy | 0.11 |
| Global lack of fund | 0.10 |
| Fear of loss of recognition | 0.08 |
| Fear of being a victim of fashion | 0.08 |
| Fear about status and role | 0.06 |

"... Also, most students do not have personal computers, and many reside in rural areas where they do not have an internet connection. Others are financially poor and thus unable to buy bundles for their phones."

Respondent 6

**Table 5. Regression coefficients for obstacles that hinder to shift to online courses teaching (N 173).**

| | B | Sig. | Exp (B) | 95% C.I. for EXP(B) | |
|---|---|---|---|---|---|
| | | | | Lower | Upper |
| Academic Qualification | | | | | |
| Diploma (Ref) | | .154 | | | |
| Bachelor | -17.071 | 1.000 | .000 | 0.000 | |
| Master | 2.336 | **.022** | 10.338 | 1.394 | 76.657 |
| PhD | .477 | .341 | 1.612 | .604 | 4.300 |
| Ever Studied Online Course | -.900 | **.081** | .406 | .148 | 1.118 |
| Knowledge on how to teach online course | 1.820 | **.004** | 6.172 | 1.777 | 21.437 |
| Knowledge and Technological Factors | 1.053 | **.017** | 2.867 | 1.210 | 6.793 |
| Environmental Factors | 1.305 | **.002** | 3.689 | 1.601 | 8.499 |
| Staff Attitudes towards teaching online courses | -1.499 | **.007** | .223 | .075 | .664 |
| Years spent in E-learning | | | | | |
| Never used E-Learning (ref) | | .002 | | | |
| 1–5 Years | -2.832 | **.000** | .059 | .012 | .281 |
| Above six years | -1.004 | **.093** | .366 | .114 | 1.182 |
| Constant | -4.031 | .010 | .018 | | |

Some respondents were concerned about quality control issues for online delivery. The respondents indicated the possibility of assessing the wrong students. The concern was how the university could be sure that the ones responding to examination questions, for example, were the intended students.

*"Again, there is a problem with quality control of the online delivery, i.e., how can we know if the one responding to the questions is the specific student we wanted to assess."*

Respondent 6

Another respondent added;

*"Whilst the quest for adopting online teaching is undoubtedly very high at this particular time, some of our students may be lacking requisite resources like Smartphones and access to the internet may be poor in the periphery. In the wake of preparing to embark on this noble teaching strategy, particularly in this era, it may be a plausible first to survey our current undergraduate and postgraduate Cohort to establish how many of them are combat-ready to seize this opportunity in the event it is introduced. Let us show what is considered huddles (Obstacles) from the Horse's Mouth, namely our customers (i.e. Current students) . . .*

Respondent 20

The views of the majority of participants suggested the need for universities to adopt online teaching despite the obstacles foreseen. The need to assess students' readiness to follow online teaching and the availability of smartphones and laptops to enhance their learning was also highlighted.

## 3.5 Proposed adaptation to online teaching by the academic staff

The last objective of the study was to document the proposed adaptation measures to online teaching by academic staff. The respondents had to write down what should be done to enable the academic staff to teach online. Apart from the obstacles highlighted, the majority of the respondent had a view that adoption of online teaching is mandatory for universities and thus, learning from other universities, such as the Open University of Tanzania, is necessary, as shared by one of the respondents;

*". . .We may borrow a leaf from the Open University of Tanzania. They have rich experiences, which can be a starting point. . .We are at the doorsteps of E-Government, E-Parliament, E-Recruitment, E-Commerce, E-Learning, et cetera. There is no going back! The issue is whether we have the right raincoats for the envisaged rains that have an array of challenges and need answers in advance to be proactive rather than reactive. As citizens of a global village, universities in Tanzania cannot afford to embrace rhetoric instead of action. . ."*

Respondent 20

Some respondents advised that universities should change the guidelines and standards for conducting online teaching before embarking on online education. This includes changing the fee structure and time frame as propounded by one respondent; *"There is a need to set up guidelines and standards on how to run online teaching and adjust payments based on the new mode of teaching"* . . . Respondent 7

Others provided the precaution that shifting to online teaching should be done with extra care; otherwise, may tarnish the image of the universities and jeopardise the quality of graduates. The specific examples mentioned are technical know-how, online course delivery and ICT infrastructures that may influence the quality of online teaching. Again, the numbers of students per instructor seem too high for many universities in Tanzania, and then the question is whether the ratio is appropriate to allow effective online interaction. This can be justified by the quotation from respondent eight, who wrote;

*"If online courses are not well designed, they may end up tarnishing the image of the University. Online courses require strong time management skills and self-motivation, a learning curve for the less technically savvy, Internet and technology-dependent susceptible to network connections, browser compatibility issues, etc., minimal in-person contact between lecturers and students might result in the production of graduates of dubious quality. Do you think you will get students who are ICT savvy*? *If not better continue with the traditional delivery model. Currently, the largest catchment of students applying to our universities is "Shule za Kata" 100% of them are computer illiterate; how can you plan for e-teaching with students with below-average computer skills"*?

Respondent 8

Another respondent added that;

*"University should provide relevant facilities and resources rather than forcing lecturers to buy and use their personal computers. The other time I used my laptop, when I had a problem that required changing windows, I could not get assistance from the relevant units. So when I see discussions about moving to e-learning, I see lecturers being forced again to find their ways from their pockets of implementing that".*

Respondent 17

These results corroborate with findings in the quantitative part that showed a relationship between ICT infrastructures, knowledge of how to conduct online courses and having an online course. Some respondents recommended that the Tanzania Commission for University provide guidelines on teaching online classes before embarking on online teaching. This should go hand in hand with changing the programme(s) curricular to capture the changes in the mode of delivery. One respondent elaborates on this;

*"I would recommend that TCU issue clear guidelines/instructions to all universities in the country on whether this should be done so that all universities could align with the TCU calendar and guidelines. Also, the same should be done by NACTE concerning non-degree students."*

Respondent 64

The qualitative findings have indicated the same trend as the quantitative, whereby the academic staff are ready to embark on online teaching; however, some preparation should be done. This preparation includes capacitating staff on how to undertake online courses, strengthening ICT infrastructure and ensuring the students have gadgets enabling them to follow online courses.

## 4. Discussion

The paper sought to determine factors that facilitate the adoption of online teaching to enhance learning while adhering to infection prevention and control measures during the outbreak of COVID-19 and another pandemic that may occur in future. The study specifically addressed three operational research questions (1) What are the factors that influence the university staff towards shifting to online teaching and (2) What are the perceived obstacles that will hinder the universities from shifting to online courses (3) What are the proposed adaptation to online courses teaching by the academic staff. The findings indicated that 88% of all respondents are willing to adopt online teaching, which is a good start for universities. However, most academic staff needed adequate university management support to enhance online teaching participation. These results support [27] who found that despite the commitment of students to follow online courses and improvement in technologies used for teaching online, the faculty still resist teaching online courses because they feel that they do not have enough support from the administration to engage in online teaching efficiently and effectively. Capacity building for faculty is significant to impart skills and knowledge in designing, developing, and instructing an online course.

The main obstacles that hinder online teaching are the need for more capacity; knowledge on how to conduct online courses; knowledge and technological factors; environmental factors; staff attitudes towards delivering online courses; and years spent using E-learning. Similarly, the study by [28] found a need for more technical expertise, insufficient orientation for learners, and a lack of release time for instructors to develop and design their online courses as barriers to faculty participation in developing and teaching online courses.

The academic staff proposed that before adopting online teaching, the universities should improve the ICT infrastructure, strengthening the capacity of academic staff to deliver online courses and ensure that all students have all requirements to enable them to follow online courses. However, different universities are in various stages regarding facilities and guidelines relevant to online learning adoption. Since the ministry responsible for education is committed and insisted the universities prepare contingency plans for online course delivery and learning, universities must start online course delivery while improving the ICT facilities.

The findings are supported by [20], who found that the determinants of online adoption in the University of Ghana are ICT infrastructure, perceived ease of use, expected benefits, organisational compatibility, and competitive pressure. This also corroborates with other studies [29–31] that show that ICT infrastructure significantly affects the adoption of online courses. These studies emphasised that for better delivery of online courses, the university should invest in ICT infrastructures while building the capacity of faculty to deliver online courses.

The study found that all academic staff know e-learning, particularly online course delivery. The willingness of academic staff to adopt online teaching is an excellent opportunity for the universities in Tanzania to invest in online teaching. This concurs with a study by [32–34], which found that human resource readiness positively influences e-learning implementation. However, a continuous, supporting environment with an internet connection and laptop availability should be prioritised to support online course delivery. In addition, the results of this study support the findings of other studies [34, 35] that indicated the main obstacles which discourage academics from online course teaching are technical problems during course preparation and course delivery and a sense of excessive mechanisation of the learning process.

Moreover, the study found that online course delivery may compromise quality issues if the universities can discover ways to provide quality education while considering the number of students to follow courses. The number of students following the course and the nature of the course requires different design and delivery techniques. This concurred with [27] that the

best practices for online course delivery need a small group of users in the virtual environment. Many users collaborating on a particular assignment may need help managing. This also reflects on the views shared by one participant that if the online course delivery needs to be better designed and managed it may end up assessing the wrong student.

## 5. Conclusion and recommendation

The study concludes that the faculty is willing to adopt online teaching. However, they felt that the university needs more support to facilitate their participation in online teaching. Therefore, the study recommends capacity building and investment in ICT infrastructures while using e-learning platforms available in their universities and accessible online teaching tools and platforms such as Google Meet. About the perceived obstacles that hinder the shifting to online courses, the study concludes that factors that include qualification, technology and staff knowledge influence the adoption of online courses. The study suggests that the universities in Tanzania strive to improve these aspects by investing in online teaching while considering the lessons learnt from the COVID-19 pandemic. Borrowing a leaf from the universities with rich experiences in online course teaching is emphasised because the world is moving to E-Government, E-Parliament, E-Recruitment, E-Commerce, E-Learning, et cetera. Due to their stake, universities are ready to adopt and run online teaching. The universities in Tanzania need to be proactive and cannot afford to embrace rhetoric instead of action. The study recommends that the universities may start offering online teaching for programmes with few students, preferably postgraduate students, and gradually extend to undergraduate students.

## Supporting information

**S1 Appendix. Reliability statistics for factors determine online courses teaching.**
(DOCX)

**S1 Data.**
(SAV)

## Acknowledgments

The authors wish to thank all those who participated in the study. We express our sincere gratitude to Mzumbe University for permitting this study to be conducted using an online platform in higher learning Institutions across Tanzania.

## Author Contributions

**Conceptualization:** Mackfallen G. Anasel.

**Writing – original draft:** Mackfallen G. Anasel.

**Writing – review & editing:** Mackfallen G. Anasel, Idda L. Swai.

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
