## [Decision Letter · Decision Letter 0]

20 Feb 2023

PONE-D-22-33260Factors determine the adoption of online teaching in Tanzania’s Universities During COVID-19 Pandemic. PLOS ONE

Dear Dr. Anasel,

Thank you for submitting your manuscript to PLOS ONE. After careful consideration, we feel that it has merit but does not fully meet PLOS ONE’s publication criteria as it currently stands. Therefore, we invite you to submit a revised version of the manuscript that addresses the points raised during the review process.

We look forward to receiving your revised manuscript.

Kind regards,

Easter Joury

Academic Editor

PLOS ONE

Journal Requirements:

"NO"

"NO"

7. Please remove your figures from within your manuscript file, leaving only the individual TIFF/EPS image files, uploaded separately. These will be automatically included in the reviewers’ PDF.

Additional Editor Comments :

In the Introduction section, please add further advantages and disadvantages of adopting online teaching / learning. 

Reviewers' comments:

Reviewer's Responses to Questions

**Comments to the Author**

1. Is the manuscript technically sound, and do the data support the conclusions?

Reviewer #1: Yes

Reviewer #2: No

2. Has the statistical analysis been performed appropriately and rigorously? 

Reviewer #1: Yes

Reviewer #2: No

3. Have the authors made all data underlying the findings in their manuscript fully available?

Reviewer #1: No

Reviewer #2: Yes

4. Is the manuscript presented in an intelligible fashion and written in standard English?

Reviewer #1: Yes

Reviewer #2: No

5. Review Comments to the Author

Reviewer #1: This manuscript presents an important issue in adopting online learning, especially in Tanzania after COVID-19, and has focused on the limitations and the reason why to adopt or not. and some important obstacles have been discussed.

The manuscript is technically sound, well-organized, and presented in an intelligible fashion.

Methods: the choice of a quantitative study design mixed with qualitative research methods is vigorous. also, the statistical methods have been performed appropriately.

The ethical approval has not been clearly stated - "The Directorate of Research, Publications, and Postgraduate Studies at Mzumbe University approved the study". it must be more specific.

The authors have shared the needed data to assess the explanation result in this manuscript. according to PLOS DATA POLICY, all data related to this manuscript must be available upon request.

The overall result and interpretation are expected for adopting online learning, it is a real problem in some low in-come-countries unfortunately.

Reviewer #2: General comments to authors

This study used questionnaire to highlight the reasons behind turning away from online teaching by the academic staff in some Tanzanian Universities. The topic is important, however, neither the study design nor the scientific writing of the manuscript achieved the minimum requirements as a scientific paper. For example, the authors did not stated the source of the questionnaire (self-developed, previous study, etc.). The information, especially the Materials and Methods and Results sections, is not organized. Was it obligatory to give lectures online or optional? What is the exact role of the supporting staff to be included in the sample?

In addition, English should be checked in the manuscript.

Detailed comments to Authors

1- Title and abstract

a- Short title: should include Tanzania.

b- The abstract does not provide some important details. For example, what is the quantitative approach? Were the participants students, lectures or professors?

2- Introduction

a- No need to make a subtitle in the introduction, or used it correctly.

3- Materials and methods

a- Where the questionnaire designed by the authors, or they used an already designed questionnaire?

b- What is the response rate? This should be clear for each participated university. The authors should give details on the number of public and private university exists, the number of the staff in each university.

c- Why the questionnaire was not distributed to students? Why it was restricted to university team?

d- The reason behind omitting one university was not clear, did you used different questionnaire for each university? Why did not they provided academic working like others?

e- No need to write the number (2) after writing it as a word (two).

f- The authors did not give any idea how the coding process was achieved.

g- Ethical consideration should be the first paragraph in Materials and methods section.

h- Please, specify why some items were deleted, what was the basis for merging some items.

i- Regarding the regression analysis, the dependent variable is clear, but what about the independent variables? The authors did not give any details on how did they code them.

4- Results

a- “79%” represents more than three-quarters rather than more than two-thirds.

b- We should know the general rate of male:female in the universities, before judging whether there was underestimation of women or not.

c- “The majority of them have master's degrees and a PhD.” What about the others’ academic degrees?

d- Have not the authors compared between the staff from ICT/Education and staff from other colleges?

e- The results section should be re-organized, to present the sample characteristics from the beginning. In addition, some details should have been added in the materials and methods section to understand the results.

f- The responses of each item of the questionnaire should be presented in the text as a table, or within the text.

6. PLOS authors have the option to publish the peer review history of their article (what does this mean?). If published, this will include your full peer review and any attached files.

Reviewer #1: No

Reviewer #2: **Yes: **Imad Barngkgei

---

## [Author Response · Author response to Decision Letter 0]

22 Jul 2023

Ref: PONE-D-22-33260

Dear, Editor-in-Chief 

We thank the editor for allowing us to revise and resubmit our paper. We would also like to thank the reviewers for their valuable remarks and are convinced that our manuscript has substantially improved with respect to the previous version. Below is the reaction to the reviewer point by point. We have made significant changes for you to track how we reacted to the second-round reviewer comments. The specific areas where we made changes are indicated in the matrix below. Our reactions to reviewers’ comments are detailed below.

General comments 

1. Is the manuscript technically sound, and do the data support the conclusions?

The manuscript must describe a technically sound scientific research piece with conclusions. Experiments must have been conducted rigorously, with appropriate controls, replication, and sample sizes. The conclusions must be drawn appropriately based on the data presented.

Reviewer #1: Yes

Reviewer #2: No

Response

Thank you for the feedback. Section 2 of the paper clearly describes the methods used for data collection, the sampling procedure and how data analysis was conducted. This enables the linkage with the results section (section 3) Before going to the conclusion, the discussion of the results was done with light of research questions to ascertain to what extent collected data answered the research questions (section 4, paragraph one). 

This approach ensures that the conclusion and recommendation made address what aspired by the data.

2. Has the statistical analysis been performed appropriately and rigorously?

Reviewer #1: Yes

Reviewer #2: No

Response

Thank you for the feedback. Although the feedback from the second reviewer is not specific on what should be improved in data analysis, we have described the data analysis similar to other studies that applied a binary model. As stated in section 2.3, paragraph two the dependent variable of this paper is the adoption of online teaching. This fall under dichotomous outcome i.e having an online course or not. With this nature, the regression analysis technique suitable for this is logistic regression/logit (see section 2.3, paragraph 2 page 7). 

For the qualitative data collected content analysis was used since it is the appropriate qualitative data analysis technique to use when you do not have a chance to probe more for additional information thus, you analyse based on the content you have (see section 2.3 paragraph 4 page 7)

In addition, the analysis section was refined and improved to ensure it was clear for the readers.

Comments

3. Have the authors made all data underlying the findings in their manuscript fully available?

The PLOS Data policy requires authors to make all data underlying the findings described in their manuscript fully available without restriction, with rare exceptions (please refer to the Data Availability Statement in the manuscript PDF file). The data should be provided as part of the manuscript or its supporting information or deposited in a public repository. For example, in addition to summary statistics, the data points behind means, medians, and variance measures should be available. If there are restrictions on publicly sharing data—e.g. participant privacy or use of data from a third party—those must be specified.

Reviewer #1: No

Reviewer #2: Yes

Responses 

The authors are ready to share the data that were used for this study. Therefore, the data used is deposited in the public repository

4. Is the manuscript presented in an intelligible fashion and written in standard English?

Reviewer #1: Yes

Reviewer #2: No

Response

The paper was reviewed and all typos and gramatical errors were corrected

Review Comments to the Author

Please provide additional details regarding participant consent. In the ethics statement in the Methods and online submission information, please specify what type you obtained (for instance, written or verbal, and if verbal, how it was documented and witnessed). 

If your study included minors, state whether you obtained consent from parents or guardians. If the need for consent was waived by the ethics committee, please include this information.

Response 

The section was improved by providing additional information on ethical issues, as indicated in section 2.4, page 8. Minors were not included in this study. In addition, written consent was given by the participants, who were informed that there was no payment or reward associated with their participation in the study. 

Reviewer #1: 

This manuscript presents an important issue in adopting online learning, especially in Tanzania after COVID-19, and has focused on the limitations and the reason why to adopt or not, and some important obstacles have been discussed.

Response

Thank you. This is so much acknowledged

Comments 

The manuscript is technically sound, well-organized, and presented in an intelligible fashion.

Response

Thank you. This is so much acknowledged

Comments 

Methods: the choice of a quantitative study design mixed with qualitative research methods is vigorous. Also, the statistical methods have been performed appropriately.

Response

Thank you. This comment is appreciated.

Comments 

The ethical approval has not been clearly stated - "The Directorate of Research, Publications, and Postgraduate Studies at Mzumbe University approved the study". it must be more specific.

Response

Thank you for the feedback. The section has been improved to clearly state the ethical approval procedures.

Comments 

The authors have shared the needed data to assess the explanation result in this manuscript. according to PLOS DATA POLICY, all data related to this manuscript must be available upon request.

Response

The data is shared as requested.

Comments 

The overall result and interpretation are expected for adopting online learning; it is a real problem in some low in-come-countries, unfortunately.

Response

Thank you for this comment. The advantage and disadvantage of adopting online teaching/learning was added. See page 4, paragraph 2.

Reviewer #2: 

Comments 

This study used questionnaire to highlight the reasons behind turning away from online teaching by the academic staff in some Tanzanian Universities. The topic is important, however, neither the study design nor the scientific writing of the manuscript achieved the minimum requirements as a scientific paper. For example, the authors did not state the source of the questionnaire (self-developed, previous study, etc.). The information, especially the Materials and Methods and Results sections, is not organized. Was it obligatory to give lectures online or optional? What is the exact role of the supporting staff to be included in the sample?

In addition, English should be checked in the manuscript.

Response

Thank you for the feedback. The following is the response which is also reflected in the manuscript.

The questions for the questionnaire were self-developed based on theoretical reviews and the conceptual model in Figure one. This is also clarified in the paper on page 6. 

Teaching online is still optional in most universities in Tanzania, except for some programmes approved to use blended delivery methods. This is clarified in the paper on page 3, paragraph 2. However, the COVID-19 pandemic forced the adoption of online teaching, especially in higher learning institutions in Tanzania, whereby most of them were relying on face–to–face, as elaborated on page 3, paragraph one. 

One of the respondent categories described in table two is supporting staff 18 (10%). The participants who were categorised as supporting staff and included in the study include the ICT officers, lab technicians and instructional designers whose role is to support lecturers and students on learning issues, therefore, their views and experience are valid for the study. 

The Language was checked as well 

Comments 

1- Title and abstract

a- short title: should include Tanzania.

b- The abstract does not provide some important details. For example, what is the quantitative approach? Were the participants students, lecturers or professors?

Response

a- The title has been revised to include Tanzania.

b- The abstract has been expanded to ensure it has a summary of the study as indicated in pages 1 and 2

Comments 

2- Introduction

a- No need to make a subtitle in the introduction or use it correctly.

Response

Thank you for the comment. The subtitle in the introduction is removed

Comments 

3- Materials and methods

a- Where the questionnaire designed by the authors, or they used an already designed questionnaire?

b- What is the response rate? This should be clear for each participated university. The authors should give details on the number of public and private university exists, the number of the staff in each university.

c- Why the questionnaire was not distributed to students? Why it was restricted to university team?

d- The reason behind omitting one university was not clear, did you used a different questionnaire for each university? Why did not they provide academic work like others?

e- No need to write the number (2) after writing it as a word (two).

f- The authors needed to give an idea how the coding process was achieved.

g- Ethical consideration should be the first paragraph in Materials and methods section.

h- Please, specify why some items were deleted, what was the basis for merging some items.

i- Regarding the regression analysis, the dependent variable is clear, but what about the independent variables? The authors did not give any details on how did they code them.

Response

a- The questions for the questionnaire were self-developed based on theoretical reviews and conceptual models in figure one. This is also clarified in the paper on page 6.

b- Section 2.2, page 5, paragraph 1 stipulates that 14 Universities were consulted, and 12 agreed to participate. Of the targeted Public Universities, were 14, and out of these, 12 universities responded. We needed to be in a position to know the total number of all staff in consulted staff. Participation in this study was voluntary, and there were no incentives attached to participation as elaborated in section 2.4, so the results in this paper indicate the true situation on the ground. 

c- The main focus of this study was to determine the perceived factors that hinder the adoption of online teaching in Tanzanian as the mitigation strategy to continue teaching despite the different pandemics as COVID-19. This is why the study focused on the views and experiences of the lecturer and supporting staff. Other studies have focused on students’ perceptions of blended learning and obstacles, and very few studies focus on online teaching. 

d- Section 2.3, paragraph 1, page 6 stipulates that 2 respondents did not respond to the concert statement, one ticked the yes box in the concert statement but did not provide any information, and one respondent was from Virginia Commonwealth University in the USA. Although the respondent from Virginia University provided relevant and helpful information related to online courses in Tanzania, it was omitted from the analysis because the study focuses on academics and supporting staff working in Tanzania Universities. The questionnaire was the same used for all Universities. 

e- Thank you for your advice. The number was removed after the word 

f- Thank you for the nice comments. The description of the coding strategy was added in section 2.3, page 7, paragraph 1.

g- Thank you for the suggestion. The paper was revised, and the ethical consideration has been included in method section 2.4

h- As highlighted in section 2.3, paragraph 2, page 7 Cronbach’s alpha coefficient values of 0.7 (70%) indicate an acceptable reliability coefficient. Therefore, the items were removed individually to ensure that the remaining items measured the same underlying construct. The improvement has been made in the text to be more precise. The items were merged based on the concepts as depicted in figure 1.

i- The dependent and independent variables are clearly shown in Figure 1, where we show Factors that determine the adoption of online teaching. In addition, the demographic factors were used as independent variables to show their contribution to online course teaching, as indicated in Table 5.

Comments 

4- Results

a- “79%” represents more than three-quarters rather than more than two-thirds.

b- We should know the general rate of male: female in the universities, before judging whether there was underestimation of women or not.

c- “The majority of them have master's degrees and a PhD.” What about the others’ academic degrees?

d- Have not the authors compared between the staff from ICT/Education and staff from other colleges?

e- The results section should be re-organized, to present the sample characteristics from the beginning. In addition, some details should have been added in the materials and methods section to understand the results.

f- The responses of each item of the questionnaire should be presented in the text as a table, or within the text.

Response

a- Thank you. We have corrected it as suggested 

b- The judgement Male:Female was done with respect to those who participated in this study. However, several studies have indicated that higher institutions in Tanzania have more Male academic staff than female ones. This was not the focus of this study.

c- As indicated in table 2, page 10, the highest qualifications start from Diploma, Bachelor, Master and PhD

d- College of affiliation was one of the independent variables entered into the regression model, with social science being the reference. Still, it did not show statistical significance, that why it was not included in the final model presented in table 5.

e- The paper was revised by presenting the sample characteristics in Table 2. The methods and results section has been improved as well.

f- Thank you for the comments. The paper was revised and included all items of the questionnaire in the paper in tables. A few items were presented in the text as examples, as this is a common practice in writing journal articles.

---

## [Decision Letter · Decision Letter 1]

12 Sep 2023

Factors Determine the Adoption of Online Teaching in Tanzania’s Universities During the COVID-19 Pandemic.

PONE-D-22-33260R1

Dear Dr. Anasel,

We’re pleased to inform you that your manuscript has been judged scientifically suitable for publication and will be formally accepted for publication once it meets all outstanding technical requirements.

Kind regards,

Eric Amankwa, Ph.D.

Academic Editor

PLOS ONE

Additional Editor Comments (optional):

Note: The title of the manuscript is missing the word "to" and should be corrected: Factors to Determine the Adoption of Online Teaching in Tanzania’s Universities During the COVID-19 Pandemic.

Reviewers' comments:

Reviewer's Responses to Questions

**Comments to the Author**

1. If the authors have adequately addressed your comments raised in a previous round of review and you feel that this manuscript is now acceptable for publication, you may indicate that here to bypass the “Comments to the Author” section, enter your conflict of interest statement in the “Confidential to Editor” section, and submit your "Accept" recommendation.

Reviewer #1: All comments have been addressed

Reviewer #3: All comments have been addressed

2. Is the manuscript technically sound, and do the data support the conclusions?

Reviewer #1: Yes

Reviewer #3: Yes

3. Has the statistical analysis been performed appropriately and rigorously? 

Reviewer #1: Yes

Reviewer #3: Yes

4. Have the authors made all data underlying the findings in their manuscript fully available?

Reviewer #1: Yes

Reviewer #3: Yes

5. Is the manuscript presented in an intelligible fashion and written in standard English?

Reviewer #1: Yes

Reviewer #3: Yes

6. Review Comments to the Author

Reviewer #1: (No Response)

Reviewer #3: The authors have addressed all the issues raised by the reviewers and formatting meet PLOSONE standard.

7. PLOS authors have the option to publish the peer review history of their article (what does this mean?). If published, this will include your full peer review and any attached files.

Reviewer #1: No

Reviewer #3: **Yes: **Kwaku Anhwere Barfi (PhD)

---

## [Editor Report · Acceptance letter]

26 Sep 2023

PONE-D-22-33260R1 

Factors Determine the Adoption of Online Teaching in Tanzania’s Universities During the COVID-19 Pandemic. 

Dear Dr. Anasel:

I'm pleased to inform you that your manuscript has been deemed suitable for publication in PLOS ONE. Congratulations! Your manuscript is now with our production department. 

Kind regards, 

on behalf of

Dr. Eric Amankwa 

Academic Editor

PLOS ONE